A simple hierarchical model for heterogeneity in the evolutionary correlation on a phylogenetic tree

http://orcid.org/0000-0003-0767-4713 Revell Liam J. 1 2 liam.revell@umb.edu
http://orcid.org/0000-0002-8331-4894 Toyama Ken S. 3
http://orcid.org/0000-0001-6483-3667 Mahler D. Luke 3
1 Department of Biology, University of Massachusetts Boston , Boston, MA , USA
2 Facultad de Ciencias, Universidad Católica de la Santísima Concepción , Concepción , Chile
3 Department of Ecology & Evolutionary Biology, University of Toronto , Toronto, Ontario , Canada
Escudero Marcial
Electronic publication date: 2022 Aug 18
Publication date: 2022
Volume: 10
Electronic Location ID: e13910
Received 2021 May 7; Accepted 2022 Jul 27
Copyright: © 2022 Revell et al.
Copyright year: 2022
Copyright holder: Revell et al.
License: This is an open access article distributed under the terms of the Creative Commons Attribution License, which permits unrestricted use, distribution, reproduction and adaptation in any medium and for any purpose provided that it is properly attributed. For attribution, the original author(s), title, publication source (PeerJ) and either DOI or URL of the article must be cited.
License URL: https://creativecommons.org/licenses/by/4.0/

Keywords: Phylogeny, Comparative methods, Model fitting

Funding: National Science Foundation DBI‐1759940 and FONDECYT, Chile 1201869 Natural Sciences and Engineering Research Council of Canada Discovery Grant RGPIN-2015-04334 This research was funded by grants from the National Science Foundation (DBI‐1759940) and FONDECYT, Chile (1201869), to Liam J. Revell; and a Natural Sciences and Engineering Research Council of Canada Discovery Grant (RGPIN-2015-04334), to D. Luke Mahler. The funders had no role in study design, data collection and analysis, decision to publish, or preparation of the manuscript.

==============================
Numerous questions in phylogenetic comparative biology revolve around the correlated evolution of two or more phenotypic traits on a phylogeny. In many cases, it may be sufficient to assume a constant value for the evolutionary correlation between characters across all the clades and branches of the tree. Under other circumstances, however, it is desirable or necessary to account for the possibility that the evolutionary correlation differs through time or in different sections of the phylogeny. Here, we present a method designed to fit a hierarchical series of models for heterogeneity in the evolutionary rates and correlation of two quantitative traits on a phylogenetic tree. We apply the method to two datasets: one for different attributes of the buccal morphology in sunfishes (Centrarchidae); and a second for overall body length and relative body depth in rock- and non-rock-dwelling South American iguanian lizards. We also examine the performance of the method for parameter estimation and model selection using a small set of numerical simulations.

Introduction

The evolutionary correlation is defined as the tendency of two phenotypic characteristics to co-evolve over evolutionary time or on a phylogenetic tree (Felsenstein, 1985; Revell & Collar, 2009; O’Meara, 2012; Caetano & Harmon, 2017; Harmon, 2019; Revell & Harmon, 2022). Many hypotheses about evolution that are tested using phylogenetic comparative methods involve the evolutionary correlation between traits. For instance, when Garland, Harvey & Ives (1992) tested the hypothesis of a correlation between phylogenetically independent contrasts (Felsenstein, 1985) for home range size and body mass in mammals, they were really asking if evolutionary increases in body size tend to be associated with increases in home range size, as well as the converse. In effect, they asked if the two traits were evolutionarily correlated. Likewise, when Ruiz-Robleto & Villar (2005) used phylogenetic contrasts to explore the relationship between relative growth rate and leaf longevity in woody plants, they were in fact investigating the tendency of these two traits to co-evolve on the phylogeny. They were measuring the evolutionary correlation between different phenotypic characteristics of plant leaves (Ruiz-Robleto & Villar, 2005).

Most analyses of the evolutionary correlation assume that the tendency of traits to co-evolve is constant over all of the branches and clades of the phylogeny. Revell & Collar (2009), however, proposed a new (at the time) likelihood-based method for testing a hypothesis of a discrete shift in the evolutionary correlation or correlations between two or more traits in certain predefined parts of the phylogeny. According to this method, which was a relatively simple multivariate extension of an important related approach by O’Meara et al. (2006; also see Thomas, Freckleton & Székely, 2006; Revell & Harmon, 2008), the rate of evolution for individual traits, and the evolutionary correlation between them, were free to vary among different regimes mapped onto the phylogeny by the user. Revell & Collar (2009) applied the method to a phylogeny and dataset for the buccal morphology of sunfishes (Centrarchidae) to test the hypothesis that the evolutionary tendency of gape width and buccal length to co-evolve was different in the highly piscivorous Micropterus clade (black bass) compared to other sunfishes.

Revell & Collar’s (2009) approach is implemented in the phytools R package (Revell, 2012) and has been applied to various questions since its original publication. For instance, Damian-Serrano, Haddock & Dunn (2021) used the method to test whether the evolutionary correlation between different aspects of the prey capture apparatus in siphonophore hydrozoans changes as a function of the type of prey they consume. The method has also been updated or adapted in different ways (e.g., Clavel, Escarguel & Merceron, 2015; Caetano & Harmon, 2017, 2019). For example, Clavel, Escarguel & Merceron (2015) developed software for modeling multivariate evolution, but with different types of constraints on the evolutionary rates or correlations between traits. Subsequently, Caetano & Harmon (2017, 2019) implemented an extension of Revell & Collar (2009) that uses Bayesian MCMC (instead of maximum likelihood) to account for several important sources of uncertainty.

The underlying model of Revell & Collar (2009) is multivariate Brownian motion (Felsenstein, 1985; Revell & Harmon, 2008; Harmon, 2019). Brownian motion is a continuous-time stochastic diffusion process in which the variance that accumulates between diverging lineages is proportional to the time since they shared a common ancestor (O’Meara et al., 2006; Revell & Harmon, 2008, 2022). The amount of covariance between species related by the tree is a direct function of the distance above the root of their most recent ancestor. At the tips of the tree, species values for a trait, x, are anticipated to have a multivariate normal distribution with a mean equal to the value at the root node of the phylogeny, and a variance-covariance matrix equal to σ2C in which C is an n × n matrix (for n total taxa) that contains the height above the root node of the common ancestor of each i, jth species pair for i ≠ j; and the total length of the tree from the root to each ith tip, otherwise (O’Meara et al., 2006; Revell, Harmon & Collar, 2008; Revell & Harmon, 2022).

In the case of multivariate Brownian motion, the diffusion process can no longer be described by a single parameter, σ2. Now, Brownian motion evolution is governed by an m × m matrix, for m traits, sometimes referred to as the evolutionary rate matrix (Revell & Harmon, 2008; Revell & Collar, 2009; Caetano & Harmon, 2017, 2019). An example of a simple, 2 × 2 Browian evolutionary rate matrix is given by Eq. (1).

(1) R=[σ12σ1,2σ2,1σ22]

In this expression, σ12 and σ22 are the instantaneous variances, or Brownian motion rates (O’Meara et al., 2006), for traits 1 and 2, respectively. Meanwhile σ1,2 (and σ2,1 – which always has the same value; i.e., R is a symmetric matrix) is the instantaneous covariance of the traits 1 and 2 (Revell & Harmon, 2008). The evolutionary correlation between traits 1 and 2, in turn, is calculated as follows.

(2) r=σ1,2σ12σ22

Alternatively then, of course, Eq. (1) can be recomposed and expressed uniquely in terms of r, σ1, and σ2.

(3) R=[σ12rσ1σ2rσ2σ1σ22]

The primary innovation of Revell & Collar (2009), as well as related methods (such as Adams, 2013; Clavel, Escarguel & Merceron, 2015; Caetano & Harmon, 2017, 2019), was to permit the instantaneous evolutionary variances and covariances of the Brownian motion process to differ in different parts of the tree that had been specified a priori by the investigator. Figure 1 shows just this type of analysis for a phylogeny of Centrarchidae (sunfishes), a discrete pair of evolutionary regimes (feeding mode: piscivorous or non-piscivorous), and a quantitative phenotypic trait dataset comprised of two different attributes of the feeding morphology: relative gape width and relative buccal length (Collar, Near & Wainwright, 2005; Revell & Collar, 2009). Note that this is a slightly different analysis from that of Revell & Collar (2009; also see Caetano & Harmon, 2019) in which the authors compared only the Micropterus clade to the rest of the phylogeny.

Figure 1 (A) Phylogeny of centrarchid fishes with feeding mode (piscivory or non-piscivory) mapped onto the edges of the tree; (B) projection of the tree in (A) into a phenotypic trait space defined by different aspects of the mouth morphology in Centrarchidae; and (C) fitted one- and two-matrix evolutionary models.

The evolutionary covariance between relative gape width and buccal length is higher in piscivorous compared to non-piscivorous fishes, and this model fits significantly better than a model in which the evolutionary covariance is assumed to be equal for the two regimes. Note that although this analysis is similar to the one that accompanied Revell & Collar (2009), here we’ve used a slightly different set of taxa and a different mapping of regimes onto the phylogeny. The phylogenetic tree is modified from Near, Bolnick & Wainwright (2005).

Figure 1A gives the phylogeny with a hypothesis about how the evolutionary regime (feeding mode) may have evolved on the tree. In this case, we arbitrarily set the ancestral regime to be non-piscivory and shift points between regimes at halfway along each edge leading to a tip or clade in the derived (piscivorous) condition; however, under many circumstances, this hypothesis could be generated using a more rigorous technique, such as stochastic character mapping (Huelsenbeck, Nielsen & Bollback, 2003). If this was done and there was a lot of variability in the stochastically mapped character histories, it would probably make sense to integrate our inference over this uncertainty (e.g., described in Revell, 2013a; see Appendix for a worked example). Figure 1B shows the phylogeny projected into the trait space. Finally, Fig. 1C shows the results of fitted Brownian evolutionary one- and two-rate matrix models (Fig. 1).

The simpler of these two models, with only one value for the evolutionary variance-covariance matrix of the Brownian process, contains a total of five parameters to be estimated: σ12 and σ22 for the two traits; σ1,2, the evolutionary covariance (or, in the equivalent reparameterization given by Eq. (3), r); and ancestral values at the root node for each trait (O’Meara et al., 2006; Hohenlohe & Arnold, 2008; Revell & Collar, 2009). By contrast, the more complex model of Fig. 1C contains a total of eight estimated parameters: σ12, σ22, σ1,2 for each of two modeled regimes (non-piscivory and piscivory), plus two ancestral states at the root.

Based on an approximately 8.1 log-likelihood unit difference between the two fitted models of our example (Fig. 1), we would conclude that the two-matrix model significantly better explains the trait data than a model in which the evolutionary rates (variances) and covariances are constant across all the branches of the phylogeny (P < 0.001; Revell & Collar, 2009). We obtain a similar result if we use information theoretic model selection (such as the Akaike Information Criterion, AIC, see below; Akaike, 1974) instead of likelihood-ratio hypothesis testing.

Looking specifically at the evolutionary correlation (r), based on Eq. (2), above, we estimate that the correlation between gape width and buccal length changes from being very slightly negative (r = −0.05) in non-piscivorous taxa, to quite strongly positive in their piscivorous kin (r = 0.80). This is consistent with stronger selection for functional integration of the different elements of the feeding apparatus in piscivorous vs. non-piscivorous lineages (Collar, Near & Wainwright, 2005; Revell & Collar, 2009).

Methods and Results

A hierarchical set of models

One obvious limitation of the approach illustrated in Fig. 1 is that it only considers two possible alternative models for the evolutionary variance-covariance matrix among traits: one in which both the evolutionary variances and the evolutionary correlation are constant; and a second in which the two mapped regimes have no similarity in evolutionary process for the two traits (Fig. 1C). In fact, it is possible to identify a number of different alternative models between these two extremes. We refer to these models as hierarchical in a similar way to how the set of sequence evolution models used for phylogeny estimation is hierarchical (Posada & Crandall, 1998): each model of increasing complexity has another simpler model as a special case.

Table 1 lists a total of eight alternative models (our original two models, from above, and six others). In square parentheses after each model, we have also provided the alphanumeric code that has been used to denominate the different models in the phytools (Revell, 2012) R software function evolvcv.lite where these models are implemented.

Table 1 Model description, model parameter estimates, log-likelihood, log(L), and AIC for one homogeneous and seven heterogeneous rate or correlation multivariate Brownian evolution models fit to the centrarchid feeding morphology evolution data of Fig. 1.

σi,j2 gives the instantaneous variance of the Brownian process (evolutionary rate) for the ith trait and jth regime. Note that this is a different use of subscripts as compared to Eq. (1) in which only traits, and not regimes, were being indicated. rj gives the evolutionary correlation between traits 1 and 2 for evolutionary regime j. In the table, regime 1 is non-piscivory and regime 2 is piscivorous feeding mode, while trait 1 is relative gape width and trait 2 is relative buccal length (Fig. 1). The best-supported model using AIC as our model selection criterion, highlighted in bold font, is model 3c: different rates for trait 2, different correlations.

Model description	σ1,12	σ1,22	σ2,12	σ2,22	r 1	r 2	log(L)	AIC	
Common rates, common correlation [1]	0.11	–	0.056	–	0.41	–	72.2	−134.4	
Different rates, common correlation [2]	0.18	0.05	0.02	0.09	0.45	–	78.0	−142.0	
Different rates (trait 1), common correlation [2b]	0.20	0.04	0.06	–	0.55	–	76.0	−140.0	
Different rates (trait 2), common correlation [2c]	0.11	–	0.02	0.10	0.33	–	75.3	−138.7	
Common rates, different correlation [3]	0.10	–	0.06	–	0.16	0.68	73.6	−135.2	
Different rates (trait 1), different correlation [3b]	0.17	0.05	0.06	–	0.36	0.65	76.5	−139.0	
Different rates (trait 2), different correlation [3c]	0.11	–	0.01	0.16	0.00	0.85	80.7	−147.4	
No common structure [4]	0.14	0.08	0.01	0.13	−0.05	0.80	81.2	−146.5	

The eight models of Table 1 are as follows: model (1) common rates, common correlation; model (2) different rates, common correlation; model (2b) different rates for trait 1 only, common correlation; model (2c) different rates for trait 2 only, common correlation; model (3) common rates, different correlation; model (3b) different rates for trait 1 only, different correlation; model (3c) different rates for trait 2 only, different correlation; finally, model (4) no common structure between the two different evolutionary variance-covariance matrices of the multivariate Brownian process.

When we analyze this complete set of models for our centrarchid dataset of Fig. 1, we find that the best fitting model (that is, the model with the highest log-likelihood) is the no common structure model in which the Brownian evolutionary variance-covariance matrix is free to differ in all possible ways depending on the mapped regime. It is, in fact, a logical necessity that model 4 has a log-likelihood that is greater than or equal to the next best model. This is because model 4, our no common structure model, has all of our other seven models as special cases. On the other hand, the best supported model (that is, the model that’s best-justified by our data taking into account model complexity; Burnham & Anderson, 2002) is model 3c (different rates in trait 2, relative buccal length, and different correlations; Table 1), indicated with bold font in the table.

Note that some other software, such as the mvMORPH R package of Clavel, Escarguel & Merceron (2015) and the ratematrix package of Caetano & Harmon (2017), also fits alternative models for multivariate Brownian evolution – such as a model in which the rate of evolution for different traits or for different regimes are constrained to be equal, a model in which the evolutionary correlation between traits, r, is constrained to be 0, or alternative matrix decompositions (such as models in which matrix “shape” or “orientation” are permitted to differ between regimes).

An empirical example: South American rock- and non-rock-dwelling lizards

In addition to the centrarchid data, above, we also applied the method to a morphological dataset of South American iguanian lizards (members of the lizard family Tropiduridae sensu lato; Toyama, 2017). For this example, we mapped habitat use of rock-dwelling vs. non-rock-dwelling (Revell et al., 2007; Goodman, Miles & Schwarzkopf, 2008) on a phylogeny of 76 lizard species. Our phylogeny was obtained from Pyron, Burbrink & Wiens (2013), but pruned to contain only the taxa of the present study, and rescaled to have a total length of 1.0. (We rescaled the tree to unit length merely so that our parameter wouldn’t need to be represented using scientific notation. Relative model fits should be completely unaffected by this rescaling.)

To set our regimes, we used a single Maximum Parsimony reconstruction of the discrete trait (rock- vs. non-rock-dwelling) on our phylogeny, in which we fixed all transitions between regimes to be located at the precise midpoint of each edge containing a state change in our reconstruction (Fig. 2). Just as in the centrarchid case, in an empirical study we would probably recommend using multiple reconstructions from a statistical method such as stochastic character mapping (Huelsenbeck, Nielsen & Bollback, 2003), and then averaging the results across stochastic maps (e.g., O’Meara et al., 2006; but see Revell, 2013a and the Discussion for some limitations associated with this common practice). This general workflow is illustrated in the Appendix.

Figure 2 Phylogenetic tree of 76 South American iguanian lizards species based on Pyron, Burbrink & Wiens (2013).

Colors indicate two different mapped ecological regimes: rock-dwelling (in black) and non-rock-dwelling (white). The tree has been rescaled to have a total depth of 1.0.

We next fit all eight of the models listed in Table 1 to a dataset consisting of body size and relative dorsoventral body depth from Toyama (2017), both calculated using geometrical definitions for size and shape (Mosimann, 1970; Klingenberg, 2016). Since rock-dwelling has previously been suggested to favor the evolution of dorsoventral flattening (e.g., Revell et al., 2007; Goodman, Miles & Schwarzkopf, 2008), we hypothesized that the evolutionary correlation between body size and depth, while generally positive across this group, could decrease or become negative in rock-dwelling forms due to ecological selection to decouple body depth from size. In Table 2, we show the parameter estimates, model fits, and Akaike weights (see section below) of all eight models from this analysis, sorted by model weight.

Table 2 Model rank, model name, model parameter estimates, log-likelihood, log(L), AIC, and Akaike weights for all eight heterogeneous rate or correlation multivariate Brownian evolution models of Table 1, fit to overall size and relative body depth in South American iguanian lizards (Fig. 2).

Column headers are as in Table 1, except for w, which indicates Akaike weight as calculated using Eq. (4).

Rank	Model	σ1,12	σ1,22	σ2,12	σ2,22	r 1	r 2	log(L)	AIC	w	
1	model 3	0.23	–	0.06	–	0.34	−0.31	55.11	−98.22	0.28	
2	model 3c	0.23	–	0.05	0.10	0.33	−0.31	56.04	−98.08	0.26	
3	model 3b	0.21	0.27	0.06	–	0.33	−0.32	55.32	−96.63	0.13	
4	model 4	0.21	0.28	0.05	0.10	0.32	−0.34	56.29	−96.58	0.12	
5	model 2c	0.22	–	0.05	0.13	0.18	–	53.90	−95.79	0.08	
6	model 2	0.20	0.30	0.05	0.13	0.20	–	54.40	−94.79	0.05	
7	model 1	0.22	–	0.06	–	0.13	–	52.31	−94.61	0.05	
8	model 2b	0.20	0.30	0.06	–	0.15	–	52.78	−93.55	0.03	

Although the weight of evidence is distributed among our top four models in the table, the most notable aspect of all of the best-supported models for these data is that they each allow the evolutionary correlation (r) to differ between the two different mapped regimes on the tree. Models that do not allow the evolutionary correlation to differ by regime (models 1, 2, 2b, and 2c from Table 1) each received less than 10% support.

We found that the evolutionary correlation between body size and size-adjusted body depth was positive in non-rock-dwelling lizards, indicating that larger lizards tended to evolve proportionally greater body depth (Table 2). By contrast, rock-dwelling forms actually exhibited a negative evolutionary correlation between body size and size-adjusted body depth. This is because larger rock-dwelling animals do not tend to evolve proportionally greater body depths. To the contrary, their size-adjusted body depth actually decreases. This is largely consistent with what’s expected given behavioral and biomechanical considerations (Revell et al., 2007; Goodman, Miles & Schwarzkopf, 2008).

A small simulation test of the method

In addition to the empirical applications given above, we tested the method using a small simulation experiment as follows. We first generated forty 100-taxon pure-birth random phylogenetic trees. On each of these trees, we simulated the history of a three-state discrete character. We rejected and repeated any simulation in which any of the three states of the trait was not observed in at least 20 tips. An example simulated tree with evolutionary regimes is given in Fig. 3A.

Figure 3 (A) Example simulated phylogenetic tree with three mapped evolutionary regimes; and (B) the phylogeny of (A) projected into a two dimensional phenotypic trait space.

The trait data in (B) were simulated under model 3b from Table 1 (different rates in trait 1, different correlations), in which the simulated evolutionary correlation between traits x1 and x2 was positive in regimes 1 and 3, but strongly negative in regime 2.

For all of the forty random trees, we simulated data under each of the eight models of Table 1. To begin each simulation, we first drew values for log⁡(σ12) and log⁡(σ22) for the two traits from a standard normal distribution (that is to say, σ12 and σ22 were randomly sampled from a log-normal distribution); and we drew a random value or values of the correlation coefficient (r) from a uniform distribution on the interval −1 to 1. Naturally, we sampled different numbers of values for log⁡(σ12), log⁡(σ22), and r depending on the model that was being used for simulation. For instance, a model with three mapped regimes (e.g., Fig. 3A) and different rates for trait 1, equal rates for trait 2, and different correlations between traits 1 and 2, would involve randomly sampling three values for log⁡(σ12), one value for log⁡(σ22), and three values for r from their respective distributions. Our simulation procedure does not fix any specific difference in the rates or evolutionary correlations between regimes. Nonetheless, it will on average result in a geometric mean ratio of the highest evolutionary rate over the lowest (for any variable σ2 simulation) of around 5.4; and a mean difference between the highest evolutionary correlation and the lowest (for any variable r simulation) of about 1.0.

An example simulated dataset generated using our procedure for different rates (trait 1), and different correlations (model 3b) is shown in Fig. 3B. In this example, we simulated the data using an evolutionary correlation between traits x1 and x2 that was positive for regimes 1 and 3, but strongly negative for regime 2 (Fig. 3B).

After completing the numerical simulations, we then proceeded to fit each of the same eight models to each simulated dataset. For each fitted model, we computed AIC and Akaike weights as follows (Akaike, 1974; Burnham & Anderson, 2002).

(4) AICi=2k−2ln⁡(li)wi=e−ΔAICi/2Σe−ΔAICj/2

Here, AICi is the value of AIC for the ith model; k is the number of parameters in the model; ln(li) is the log-likelihood of the ith model; and ΔAICi is the difference in AIC between the ith model and the model with the minimum AIC score in the set. In general, we should prefer the model with the lowest overall value for AIC, and can interpret the Akaike model weights (w), from Eq. (4), as a measure of the strength of evidence supporting each of the models in our set (Akaike, 1974; Wagenmakers & Farrell, 2004).

After fitting all eight models to each of our 40 × 8 = 320 simulated datasets, we next simply calculated the fraction of times in which the generating model was selected as the ‘best’ model (as well as second best, third best, and so on). These results are summarized in Table 3. In general, we found that the generating model tended to be selected as the best or second best model over 86% of the time in simulation, under the simulation conditions described above (Table 3).

Table 3 Model name and the fraction of times from 40 simulations in which the generating model (in rows) was identified as the best, 2nd best, 3rd best, or worse than 3rd best model using AIC model selection.

Model name	Best	2nd best	3rd best	≥4th	
model 1	0.65	0.12	0.05	0.17	
model 2	0.70	0.17	0.10	0.02	
model 2b	0.70	0.05	0.15	0.10	
model 2c	0.55	0.20	0.15	0.10	
model 3	0.75	0.17	0.02	0.05	
model 3b	0.75	0.10	0.10	0.05	
model 3c	0.52	0.30	0.12	0.05	
model 4	0.65	0.35	0.00	0.00	

In addition, we also calculated the average weight ( w¯) of each of the forty datasets for each model. These results are summarized in Fig. 4. This analysis shows that the generating model (in rows) also tended to have the highest average Akaike model weight (in columns; Fig. 4).

Figure 4 Mean Akaike weight for all eight models (in columns) for each of the eight generating models (in rows).

Simulation conditions were as described in the text.

For each generating model, the next highest average Akaike model weights tended to be observed in models of similar complexity. For instance, when the generating model was model 4 (no common structure), we found the highest average model weight for model 4 (0.65); and then the next highest average model weights for models 3b (different rates for trait 1, different correlations; 0.14) and 3c (different rates for trait 2, different correlations; 0.13). Likewise, when the generating model was model 1 (common rates, common correlation), we found the highest average model weight for model 1 (0.37), and the next highest average model weight for model 3 (common rates, different correlation; 0.14).

Lastly, having established that the method tends to select the generating model used for simulation, we also proceeded to assess parameter estimation. Here, we computed both the correlation between the generating and estimated values for each model (to measure precision); and the mean difference (to measure accuracy or bias). Correlations for all parameters and all models are given in Table 4. Table 5 shows the mean difference between the generating and estimated parameter values for each model.

Table 4 Model name and correlation between the generating and estimated parameter values (rounded to two digits) for each model.

For σ2 both the generating and estimated values were transformed to a log scale before computing the correlation. Simulation conditions were as described in the text and illustrated in Fig. 3. In all cases, the generating model was used for estimation.

Model	r 1	r 2	r 3	σ1,12	σ1,22	σ1,32	σ2,12	σ2,22	σ2,32	
model 1	0.99	–	–	0.99	–	–	0.99	–	–	
model 2	0.99	–	–	0.95	0.95	0.97	0.93	0.96	0.97	
model 2b	0.99	–	–	0.96	0.97	0.97	0.98	–	–	
model 2c	1.00	–	–	0.99	–	–	0.97	0.94	0.94	
model 3	0.98	0.96	0.98	0.99	–	–	0.99	–	–	
model 3b	0.98	0.94	0.95	0.98	0.96	0.98	0.99	–	–	
model 3c	0.98	0.95	0.96	0.99	–	–	0.98	0.95	0.96	
model 4	0.97	0.96	0.97	0.97	0.97	0.97	0.94	0.96	0.96	

Table 5 Model name and mean difference between the generating and estimated parameter values (rounded to two digits) for each model.

Just as in Table 4, for σ2 both the generating and estimated values were transformed to a log scale before computing the differences. As such the reported differences are proportional to the generating values. (That is to say, a log difference of 0.04, for example, indicates a mean error of e0.04 or ∼4.1%.) Simulation conditions were as described in the text and illustrated in Fig. 3. In all cases, the generating model was used for estimation.

Model	r 1	r 2	r 3	σ1,12	σ1,22	σ1,32	σ2,12	σ2,22	σ2,32	
model 1	−0.01	–	–	0.04	–	–	−0.01	–	–	
model 2	−0.03	–	–	0.10	0.07	0.02	0.10	0.09	0.02	
model 2b	−0.00	–	–	0.10	0.02	0.06	0.02	–	–	
model 2c	−0.01	–	–	0.02	–	–	0.02	−0.02	0.09	
model 3	0.01	0.02	−0.03	−0.00	–	–	0.01	–	–	
model 3b	−0.02	0.01	0.03	0.02	0.03	0.01	0.03	–	–	
model 3c	−0.00	0.03	0.03	0.04	–	–	−0.01	0.05	0.07	
model 4	−0.02	−0.04	−0.01	0.04	0.07	0.11	0.07	0.11	0.07	

Notes on implementation

The model and methods of this study have been implemented for the R statistical computing environment (R Core Team, 2021), and all simulations and analyses for this study were done using R.

The method that we describe in the article is implemented as the function evolvcv.lite of the phytools R package (Revell, 2012). phytools in turn depends on the core phylogenetics R packages ape (Paradis & Schliep, 2019) and phangorn (Schliep, 2011), as well as on a number of other R libraries (Venables & Ripley, 2002; Ligges & Mächler, 2003; Lemon, 2006; Plummer et al., 2006; Chasalow, 2012; Becker et al., 2018; Gilbert & Varadhan, 2019; Azzalini & Genz, 2020; Qiu & Joe, 2020; Warnes, Bolker & Lumley, 2020; Goulet et al., 2021; Pinheiro et al., 2021).

Though the simulations and empirical analyses presented herein use only two or three mapped regimes, the current implementation of this method in phytools, evolvcv.lite, permits an unlimited number of mapped regimes. The reader should keep in mind, however, that the number of parameters to be estimated will rise in direct proportion to the number of regimes.

Discussion

The evolutionary correlation is defined as the tendency of two different phenotypic traits to co-evolve (Harmon, 2019; Revell & Harmon, 2022). Traits are said to have a positive evolutionary correlation if a large increase in the value of one trait tends to be a accompanied by a similarly large increase in the second, and vice versa. Traits can be evolutionarily correlated for a wide variety of reasons. For instance, a genetic correlation between traits, if persistent over macroevolutionary time periods, will tend to cause two phenotypic characteristics to evolve in a correlated fashion, even under genetic drift (Schluter, 1996; Blows & Hoffmann, 2005; Hohenlohe & Arnold, 2008; Revell & Harmon, 2008). Genetic correlations between traits in turn tend to be caused by pleiotropy, such as when one quantitative trait locus affects the expressed value of two different phenotypic attributes (e.g., Gardner & Latta, 2007).

More often, however, when an evolutionary correlation between traits is observed, natural selection tends to be purported. For instance, the evolutionary correlation between water-related plant traits, such as pseudobulb length and stomatal volume, observed by Sun et al. (2020), was interpreted by the authors as evidence for natural selection acting to favor certain combinations of trait values over others. Likewise, when Goodwillie et al. (2009) found an evolutionary correlation between reproductive outcrossing rate and the product of flower number and size in plants, they hypothesized that this was due to selection favoring increased investment in structures to attract pollinators in outcrossing compared to selfing taxa. Numerous questions in evolutionary research involve measuring the evolutionary correlations between traits (Felsenstein, 1985; Harmon, 2019), and in many cases it may be sufficient to fit a single value of the evolutionary correlation between characters for all the branches and nodes of the phylogeny. Under other circumstances, however, it is useful or necessary to permit the evolutionary correlation to assume different values in different parts of the tree.

For example, in the present study we used data for centrarchid fishes to test whether feeding mode influences the evolutionary tendency of two different aspects of the buccal morphology to co-evolve (Revell & Collar, 2009). We hypothesized that natural selection for functional integration of the feeding apparatus constrains different buccal traits to evolve in a coordinated fashion in piscivorous lineages, but not in their non-piscivorous kin (Collar, Near & Wainwright, 2005). Indeed, although we used a slightly different dataset and phylogeny here (focusing on all piscivorous centrarchids, rather than just the Micropterus clade), our analysis largely reiterates the finding of Revell & Collar (2009) in showing that a model with different evolutionary correlations between traits depending on feeding mode significantly better explains our morphological trait data, compared to a model in which the evolutionary correlation is forced to have a constant value across all the branches of the phylogeny. Like Revell & Collar (2009), we also found that the evolutionary correlation between buccal traits is high and positive in piscivorous but not non-piscivorous lineages (Table 1). Unlike Revell & Collar (2009), however, we found that the best-supported model was one in which the evolutionary rate (σ2) for buccal length, but not gape width, was also free to differ in different parts of the phylogeny.

Likewise, we present data for the evolution of overall body size and size-adjusted dorsoventral body depth in South American iguanian rock-dwelling and non-rock-dwelling lizards, a group rich in habitat transitions (Fig. 2; Toyama, 2017). Based on prior research (Revell et al., 2007; Goodman, Miles & Schwarzkopf, 2008), we hypothesized that selection might favor the decoupling of a normally positive evolutionary correlation between the two traits to permit the evolution of greater dorsoventral compression in rock-dwelling species. Indeed, all four of the best-fitting models in our analysis were ones in which the evolutionary correlation was permitted to differ by habitat use, rock or non-rock—and the evolutionary correlation differed between regimes in exactly the predicted direction (Table 2). Models where the evolutionary rates (σ2), but not the evolutionary correlation (r), differed across the tree received very little support.

Finally, we undertook a small simulation study of our method. We found that the generating model in simulation also tended to be the model that was most often chosen via our model selection procedure (Table 3; Fig. 4). When the generating model was not best-supported, a model of similar parameterization tended to be selected instead (Fig. 4). We also showed that the accuracy and precision of parameter estimation was reasonably high (Tables 4 and 5).

Table A1 Log-likelihoods and model support for three alternative models for discrete character evolution of non-rock- or rock-dwelling in tropidurid lizards, assuming non-rock-dwelling as the ancestral condition.

ER: equal-rates model, assumes backward and forward transitions between habitat types occur at the same rate. ARD: all-rates different, permits different forward and backward transition rates. Directional: assumes that evolution invariably proceeds from non-rock-dwelling to rock-dwelling, and never the reverse. Even after accounting for model complexity. ARD is the best-supported of these three models.

Model	log(L)	d.f.	AIC	
ER	−43.37	1	88.75	
ARD	−41.27	2	86.55	
Directional	−46.25	1	94.51	

Limitations

Although we believe that the method of this article extends existing methodology in an interesting way, it comes with its own limitations.

The most evident limitation of the approach described herein and implemented in the phytools evolvcv.lite function is that it is restricted to analyzing no more than two quantitative traits at a time. This practical constraint arises from the fact that we have chosen to parameterize our set of nested models using a common (or different) correlation and common (or different) evolutionary rates paradigm. This paradigm does not extend as easily to more than two traits. The reasons for this are entirely pragmatic. With only two quantitative traits, the likelihood function is defined (and can thus be evaluated) for any values of σ12 and σ22>0 in each regime, and for any −1 < r < 1. For three or more quantitative traits, on the other hand, many combinations of the correlation coefficients r1,2, r1,3, and so on, can lead to an evolutionary covariance matrix that is uninvertible (non-positive definite), and thus a likelihood that cannot be computed. This is a solvable problem and one of us [LJR] is presently working to extend evolvcv.lite to an arbitrary number of continuous characters; however, this has required implementing numerical optimization with dynamic bounds and has thus far proved challenging. Ultimately, we hope that this functionality will be added to the phytools R package at some time in the not too distant future.

Likewise, evolvcv.lite only includes multivariate Brownian motion as a model of trait evolution. Expanding beyond Brownian motion, for instance to include the Ornstein-Uhlenbeck (OU) model (e.g., Butler & King, 2004; Beaulieu et al., 2012), is a highly interesting future direction. Even more so than extending to more than two traits, this addition would present a number of additional challenges and is thus beyond the scope of our present work. Not least among these is the substantial increase in model complexity that a multi-regime multivariate OU model would entail. (In particular, each of σ and α, the stochastic and stabilizing selection or ‘rubber band’ parameter in the OU model, respectively, would require a separate m × m matrix for each regime.)

Finally, our approach requires that different regimes on the tree are hypothesized a priori by the investigator. In this way, it follows a paradigm established in Butler & King (2004) and O’Meara et al. (2006), and re-described in a variety of other places since (e.g., Revell, 2008; Beaulieu et al., 2012; Revell et al., 2021; and others). An alternative approach would be to, for example, use reversible-jump Markov Chain Monte Carlo (rjMCMC) to allow our data to inform the location of regime shifts on the tree (e.g., Eastman et al., 2011; Uyeda & Harmon, 2014; also see Revell, 2021 for a different approach using penalized likelihood). In fact, you could go one step further still and sample both the number and position of regime shifts, and model complexity (as given in Table 1) from their joint posterior probability distribution. We feel that this is certainly an intriguing future direction, but out of scope for this relatively modest contribution.

Other considerations

In the current article we have focused on our model for quantitative trait evolution and have left consideration about how regimes are obtained or set on the tree mostly aside. It is entirely appropriate to use this method for circumstances in which there is no ambiguity about the mapping of regimes onto the edges and clades of the tree, such as, for instance, when fitting a model with heterogeneous evolutionary correlations between monophyletic groups or across difference time periods. Often, however, investigators may be inclined to use our approach to study the influence of a mapped discrete character on the evolutionary process for their quantitative traits. In that case, we would recommend considering an approach that allows the explicit incorporation of uncertainty, such as integrating over a set of character histories sampled in proportion to their probability under a model (Huelsenbeck, Nielsen & Bollback, 2003; e.g., Price, Friedman & Wainwright, 2015), as illustrated in the Appendix of this article for our rock- and non-rock-dwelling lizard empirical example. As shown in Revell (2013a), however, this common practice can also result in certain biases, such as causing estimated evolutionary rates to resemble each other more closely than the generating rates (Revell, 2013a; also see May & Moore, 2019). An interesting approach that was suggested by Caetano & Harmon (2019) involves first generating a posterior sample of discrete character histories using stochastic mapping, and then sampling from this set during Bayesian MCMC. This seems worthy of further examination.

In addition, herein we have focused on model selection under simulation conditions in which the true, generating model is invariably included among the set of models we fit to our data. Under these conditions, we tended quite strongly to select the correct model, or if not the correct model, one very similar to it (Table 3; Fig. 4). We do not explicitly consider the inevitable condition in which the true model is not among those of our model set. For other types of comparative methods, model inadequacy can result in inference errors that are quite grave (e.g., Rabosky & Goldberg, 2015; Beaulieu & O’Meara, 2016). Indeed, the general problem of model adequacy is underappreciated in phylogenetic comparative methods (Boettiger, Coop & Ralph, 2012). Some promising approaches to the problem have already been developed (e.g., Pennell et al., 2015; Uyeda, Zenil-Ferguson & Pennell, 2018; Duchene et al., 2018) and we recommend that it be the subject of continued research.

Lastly, we have chosen to concentrate on using maximum likelihood for estimation. In fact, it would be entirely conceivable to take a Bayesian MCMC approach (e.g., following Caetano & Harmon, 2017, 2019), or perhaps even use rjMCMC to integrate over the different models of Table 1 in proportion to their posterior probability, as described in the previous subsection. One advantage of this approach is that it provides a natural framework within which to incorporate uncertainty about the fitted model and its parameters. One disadvantage is that it requires us to decide on appropriate prior distributions for our model parameters, evaluate convergence of the MCMC to the posterior, and summarize the posterior sample (Caetano & Harmon, 2019). It does not obviate any of the other limitations or considerations detailed above.

Relationship to existing methods

The method of this paper is an extension of an existing model described in Revell & Collar (2009). In that article, the authors describe an approach to analyzing heterogeneity in the evolutionary correlation, Brownian rate, and Brownian covariance between different pre-specified branches or clades of a phylogenetic tree. In Revell & Collar (2009) the authors compare two alternative models: one in which evolutionary rates and the evolutionary correlation between characters are free to differ between regimes; and a second in which they are not. Here, we add a set of six additional intermediate models (for two quantitative traits) in which regimes are permitted to share various aspects of their evolutionary process in common (rates for one character or another, evolutionary correlation), while differing in others. This method was also extended to a Bayesian framework by Caetano & Harmon (2017, 2019), and as such the work we present here is also closely related to this research.

Readers of this article who are familiar with phylogenetic comparative methods might observe that it’s also possible to model multivariate trait evolution in which the relationship between traits changes as a function of a discrete factor using a phylogenetic generalized analysis of covariance (Grafen, 1989; Rohlf, 2001; Revell, 2010; Mundry, 2014; Fuentes-G et al., 2016). In this case, we would simply fit a linear model in which a single dependent variable (y) varied as a function of a discrete factor (the tip regime), a continuous variable (x), and their interaction (to permit differences in slope between regimes), while assuming that the residual error in y has a multivariate normal distribution given by the structure of the tree (Rohlf, 2001; Revell, 2010; Fuentes-G et al., 2016). Indeed, this is a valid approach for asking how the relationship between traits changes among lineages of a reconstructed phylogeny. We nonetheless feel that our method adds value for many investigators because it permits an arbitrary (not just tip) mapping of discrete regimes, because it doesn’t require the user to specify dependent and independent variables in the model, because it easily allows us to take into account sampling error in the estimation of species’ means (following Ives, Midford & Garland, 2007), because it’s readily extensible to more than two traits whose correlations might also be expected to change as a function of the mapped regimes, and, finally, because it’s more directly connected to a hypothesized evolutionary process for the traits on our phylogeny (Hohenlohe & Arnold, 2008; Revell & Harmon, 2008).

Conclusions

The evolutionary correlation is defined as the tendency for changes in one phenotypic attribute to be associated (positively or negatively) with changes in a second trait through evolutionary time or on a phylogenetic tree (Harmon, 2019; Revell & Harmon, 2022). Many questions in phylogenetic comparative biology involve measuring the evolutionary correlations between characters using phylogenies. Often, it’s sufficient to assume a constant value of this evolutionary correlation through time or among clades. Here, however, we present a hierarchical series of models in which we permit the rate of evolution for traits, and their evolutionary correlation, to differ in different parts of the phylogeny that have been specified a priori by the investigator.

Appendix

In the main text of this article, we intentionally focused on fitting heterogeneous correlational trait evolution model in which regimes for the evolutionary correlation were assumed to be ‘known’ a priori by the investigator.

Although this may sometimes be the case (for instance, in a study in which the regime is based on membership to a specific clade, or in which the reconstructed history of a discrete character is unambiguous), quite often our hypothesized regimes will be based on the unknown history of a discrete character. In that case, a common practice is to first sample plausible histories in proportion to their probability using a statistical procedure called stochastic character mapping (Huelsenbeck, Nielsen & Bollback, 2003), fit the model to each history, and then average over this set (e.g., Price, Friedman & Wainwright, 2015).

Here, we illustrate this workflow using rock and non-rock dwelling in the tropidurid lizards. To do this, we began by fitting a set of three, alternative discrete trait evolution models: an equal-rates model (ER); an all-rates-different model (ARD); and a directional model, in which evolution of the discrete character is allowed to proceed from non-rock to rock dwelling, but not the reverse (Revell & Harmon, 2022). We found that the best-supported model was the ARD model (Table A1).

We next sampled 100 stochastically mapped character histories in proportion to their probability under our best-fitting model using the method of Huelsenbeck, Nielsen & Bollback (2003), Bollback (2006) as implemented in the phytools R package (Revell, 2012). Figure A1 shows (in A) a single, representative, haphazardly selected stochastic character history of our rock-dwelling vs. non-rock-dwelling; and (in B) a continuous visualization of the posterior probability that each edge of the tree was in each of the two states (following Revell, 2013b), under our hypothesized discrete character evolution model.

Figure A1 (A) Example stochastic character map for rock dwelling (vs. non-rock-dwelling) in tropidurid lizards; and (B) posterior probability of rock-dwelling from 100 stochastic character histories.

With 100 stochastic character mapped trees in hand, we then fit all of our eight models to each tree. Our findings were highly congruent with what we showed in Table 2 of the main text. In particular, for 80% of stochastic character mapped trees, model 3 (common rates, different correlation; Table 1) was the best-supported model (Table A2). Likewise, across 96% of all stochastic character mapped histories, either model 3 or model 3c (different rates for character 2, relative body dorsoventral, different correlation; Table 1) was the best-supported model (Table A2).

Table A2 Frequency across all stochastic mapped trees with which each model ranked 1st, 2nd, 3rd, and so on.

w¯ gives the mean Akaike model weights of each model, averaged across all stochastically mapped trees. Models are as in Table 1.

Rank	Model 1	Model 2	Model 2b	Model 2c	Model 3	Model 3b	Model 3c	Model 4	
1st	0.01	0.01	0.00	0.02	0.73	0.00	0.23	0.00	
2nd	0.00	0.02	0.00	0.09	0.09	0.20	0.54	0.06	
3rd	0.03	0.07	0.00	0.11	0.04	0.46	0.19	0.10	
4th	0.22	0.03	0.00	0.10	0.01	0.06	0.04	0.54	
5th	0.33	0.08	0.05	0.17	0.12	0.05	0.00	0.20	
6th	0.22	0.03	0.18	0.33	0.00	0.19	0.00	0.05	
7th	0.18	0.22	0.36	0.18	0.00	0.03	0.00	0.03	
8th	0.01	0.54	0.41	0.00	0.01	0.01	0.00	0.02	
w¯	0.06	0.05	0.03	0.08	0.32	0.13	0.22	0.10	

Lastly, we computed model-averaged parameter estimates under each of the top four ranked based on a rank-choice vote (most common best, second best, and so on; Table A2), in which model averaging was done across all 100 stochastic character histories (Fig. A1). This is similar to a simple average, but up-weights stochastic maps that make the observed data for our quantitative traits more probable (under the model). The results from this analysis are given in Table A3.

Table A3 Model-averaged parameter estimates for the top four ranked models.

Model-averaging was computed for a given model, across stochastic character maps.

Rank	Model description	σ1,12	σ1,22	σ2,12	σ2,22	r 1	r 2	
1	common rates, different correlation [3]	0.22	–	0.06	–	0.38	−0.31	
2	different rates (trait 2), different correlation [3c]	0.22	–	0.05	0.09	0.35	−0.24	
3	different rates (trait 1), different correlation [3b]	0.22	0.24	0.06	–	0.37	−0.31	
4	no common structure [4]	0.21	0.27	0.05	0.09	0.35	−0.23	

In addition, it’s possible to use stochastic character mapping to obtain a measurement of our degree of uncertainty in parameter estimation that is due to ambiguity in the discrete character history. In this case, one would just compute a variance (or a weighted variance, using Akaike weights) in each estimated parameter across the set of discrete character histories. This measure of uncertainty in the values of estimated parameters does not, however, include estimation error from any given character history, which must also be measured (for instance, by using the Hessian matrix of second order partial derivatives of the likelihood surface; e.g., following Price, Friedman & Wainwright, 2015). We do not show this here, but it is relatively straightforward to accomplish in R.

Additional Information and Declarations

Competing Interests

Author Contributions

Data Availability

The authors declare that they have no competing interests.

Liam J. Revell conceived and designed the experiments, performed the experiments, analyzed the data, prepared figures and/or tables, authored or reviewed drafts of the article, and approved the final draft.

Ken S. Toyama conceived and designed the experiments, performed the experiments, analyzed the data, prepared figures and/or tables, authored or reviewed drafts of the article, and approved the final draft.

D. Luke Mahler conceived and designed the experiments, analyzed the data, prepared figures and/or tables, authored or reviewed drafts of the article, and approved the final draft.

The following information was supplied regarding data availability:

The raw data and code is available at GitHub: https://github.com/liamrevell/evolvcv.lite.figures/.

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
