# Peer review of "A simple hierarchical model for heterogeneity in the evolutionary correlation on a phylogenetic tree"

_PeerJ, doi:10.7717/peerj.13910_

## Round 0.1 · original submission · Minor Revisions

The two reviewers agree that this may be a significant contribution phylogenetic comparative methods. They also agree that the manuscript could be improved. Please, consider the suggestions by the two reviewers and re-submit a new improved version. Thanks!

·

Basic reporting

The writing of this manuscript is clear and unambiguous. There are certain grammatical edits that I have made which can be found in the line edits below. The authors do use many conjunctions, which I recommend should be removed.

The manuscript is well cited.

The authors supply all of the figures and tables in the main text. I am unaware if they are making the raw data available, but this would be a nice addition.

The manuscript is very self-contained and the authors do not stray far from their topic.

Experimental design

This project fills a significant gap in the methodology used to examine evolutionary correlations.

The research question is clear, well defined, and relevant.

The examination and testing of the implemented method is rigorous and includes multiple empirical tests and one simulated test. I do however, believe that the authors should increase their simulations for Table 3, per my line edits.

The methods are described in sufficient detail, if R code is supplied then this should be able to be easily replicated.

Validity of the findings

The findings of this manuscript validate the use of the implemented method: evolvcv.lite.

As I believe they have, the authors should include all underlying data and R documentation.

The conclusions are well stated and linked to the original topic of the manuscript. No significant speculation or deviation from the main topic was evident.

Additional comments

This is a well written, well cited, and nicely executed project that adds a useful method to the phytools package. I believe that the testing of this method was robust both in their empirical and simulated examinations. I think that this method will be utilized by researchers interested in testing the evolutionary correlation between traits under different regimes across the phylogeny.
I do have a few questions that the authors should address in the text. 1. How many discrete traits (regimes) can be implemented in evolvcv.lite? 2. Why do the authors only include a Brownian motion model and not an OU model as well? Of course this would add another dimension to this model and project, but this may be of interest to users. 3. What occurs in the situation where there are different rates or evolutionary correlations under the same regime (discrete trait) that are in different parts of the phylogeny? For instance, in Fig. 3a, if the clade in regime 3 at the lower end of the phylogeny was actually evolving under a faster rate than the clade in regime 3 at the top of the phylogeny, would we be able to pick that up? 4. This relates to my 3rd point, is there a way to allow the rate and evolutionary correlation to vary across the tree without being correlated to a specific regime (i.e., not specified a priori)?
If my comments and edits are addressed, I believe this manuscript should be accepted for publication with minor revisions.



Line edits and other minor edits

Minor comment: I recommend removing all conjunctions and spelling out each word.

28: “Lots” may be too informal, I would recommend changing this to “Many”, or if the authors believe this is closure to half or the majority, they could use a more indicator

35-37: I recommend making the final sentence of this paragraph a more general statement about evolutionary correlations. For instance it could read something like, “In many instances researchers measure the evolutionary correlation between different phenotypic traits across the phylogeny.”

Figure 1: Great figure, clearly articulated, can you indicate if fig. 1a was generated using stochastic character mapping or not? I assume so but, I think it would be good for the reader.

Line 107: I think this is a fantastic implementation of heterogeneity in evolutionary correlation on the tree and I recommend adding some sort of concluding paragraph on the importance of this newly implemented method.

Table 1: punctuation seems a bit off on the 4th and 5th line, I recommend removing the parentheses from the parenthetical statement

164: I recommend adding models 1-2b to Table 1 for the readers reference?

Table 3: Is there a reason that when model 3c was the generating model it was only chosen 40% of the time as the best model? Also, why were there only 20 simulations? Perhaps this could be increased.

224: I recommend removing the word “important”, this is subject to the field of the researcher using R.

240: I recommend the authors be more specific with when they indicate “water-related plant traits”
260-261: I recommend including a sentence or two on why the atuhros think their model different from their previous one in Revell and Colloar 2009

Sincerely,
Jacob Suissa

Reviewer 2 ·

Basic reporting

The article is clearly written and the ideas have been structured and presented in a excellent order and fluidity. The article is pleasant to read and the author’s arguments are easy to follow.

The article is lacking a couple of relevant references that I have described, and justified, later in this review. Besides that, the introduction is adequate. It could be more detailed in terms of the role of evolutionary correlations in shaping the biodiversity as a whole, but it is not necessary.

The figure of the article are clear and in adequate number. Figure 4 could be significantly reduced in size without any detriment to its quality or information content. Some legends, such as the rightmost vertical numbering of Akaike weights in Figure 4 could use larger font sizes.

It is not clear if the raw data for the empirical and simulation examples has been made available. Authors should make it clear if the raw data is available and where it can found.

Experimental design

The article use model comparison using AIC and AIC weights to rank multiple alternative models of trait evolution that are able to estimate changes in the evolutionary correlations among continuous traits. The authors used simulations to test the performance of the approach and empirical data to show a couple of examples.

The authors suggest in the text that one should take into account the uncertainty associated with the predictive regimes used to infer the location of the multiple multivariate BM matrices fitted to the tree. However, the study does not implement this recommendation. This is contradictory and, following the author own arguments, decrease the overall confidence of the results.

The authors focus solely on model comparison using AIC and there is no discussion about parameter estimates. Parameter estimates are fundamental to the use of models to understand evolutionary patterns. Even if one particular model has the best fit to the data, there is no guarantee that this is an adequate model (see general discussion in Rabosky and Goldberg 2015 – Systematic Biology https://doi.org/10.1093/sysbio/syu131; although this reference is about a very different model and question, their point about model adequacy is relevant to most phylogenetic comparative models). Parameter estimates are also important to draw conclusions about the biology and evolutionary history of the group. For example, the best model in a set can estimate evolutionary correlations to be very distinct from the generating model. How, and how often, the parameter estimates drive us to arrive in wrong conclusions about the study system? Here the authors do not include any test, mention, or discussion about the role of parameter estimates in conducting the study of evolution using stochastic models.

Validity of the findings

Several parts of this study can be improved, but in overall the findings reported here are supported by the data. It is important to note that a previous study has criticized some of the aspects of the analyses used here (Caetano and Harmon 2019 – Systematic Biology https://doi.org/10.1093/sysbio/syy067) and the authors should address this here.

Additional comments

Here the authors used AIC and AIC weights to rank models that increase in complexity from a single rate fit to the whole tree to differences in the individual rates of trait evolution and evolutionary correlations. Model comparison using AIC and AIC weights is widely used in the literature. The models the authors use here, with each of the elements of the matrix free to vary and all the special cases of this more general model, have been available before in multiple R packages: “phytools”, “mvMORPH”, and “ratematrix”. Thus, it is not clear to me why the authors have framed this study as a “hierarchical model” or “method”. Yes, the increasingly more complex relationship of the models tested here are hierarchical, but this is not far from standard practice when using AIC to compare among models. The general philosophy behind the use of AIC is to erect a pool of models that can potentially explain the patterns in the data that we are interested in. Then the AIC, and the AIC weights, help us to select which of these models best explain the patterns observed in the data. In this sense, many, if not most, applications of model comparison in phylogenetics are “hierarchical models”. For instance, the standard practice of choosing models of sequence evolution for phylogenetic estimation use a principle very similar to the one applied here. In summary, I would like to know the reasoning behind the author’s decision to describe this study as something “new”, as it is my impression from the title as well as the language used in the main text.

Line 50: “the the”.

Lines 95 to 105: Caetano and Harmon (2019 – Systematic Biology) discuss about the uncertainty associated with multivariate BM models and use MCMC to estimate the parameters of the model in a similar example also using Centrarchidae fishes. What is the impact of uncertainty in these results?

Lines 134 to 137: Caetano and Harmon (2017 – MEE and 2019 – Systematic Biology) introduces a test using summary statistics computed from the posterior distribution of the parameter estimates that allows one for evaluating evidence for shifts in the evolutionary correlation and the rates of trait evolution for each one of the traits in the model. According to them, the test can detect many of the shifts listed on Table 1. How the approach implemented on the “ratematrix” package compares to the one described here?

Note on the model implementation: Caetano and Harmon (2019 – Systematic Biology) describe an adaptation of the pruning algorithm that can be applied to multivariate Brownian Motion models. Have the authors considered this algorithm?

Note on incorporating uncertainty: May and Moore (2019 – Systematic Biology https://doi.org/10.1093/sysbio/syz069) argues that one need to consider a more complex model that relax rates of trait evolution across the branches of the tree instead of the single rate regime when trying to incorporate rate heterogeneity in the analyses. Their argument is that, similar to the problem of model inadequacy in BiSSE that has been proposed to be solved using HiSSE, one can falsely accept the two regime evolutionary rate matrix because it captures more of the rate variation whereas the most adequate model might be one as complex (or more complex) that do not map to the a priori regime erected by the researcher. How this general criticism apply to the approach described here?

Lines 148 to 152: If this is the recommendation from the authors, why has this approach not been applied here? The package “phytools” implement stochastic mappings, so I don’t see what is the impediment here. It is important to note that method description publications are very often later used as the standard approach empiricists will adopt. If the authors have a recommendation about the correct (or best) practice to do the analysis, please let the publication be the example that other researchers and students can follow. This is also important because students will read the empirical examples in search of the interpretation of the results, which can be challenging for complex approaches.

Notes on the simulation test: Previous studies have shown that it can be hard to accurately estimate parameters as the number of dimensions in a multivariate model increases. Can the authors comment/discuss the certainty in parameter estimation of these models?

Are the simulation replicates using the exact same regime mapping used to generate the data? In empirical situations the researcher might not be able to identify the true underlying regimes that map to the correct point in the evolutionary history of the clade that the shift have happened (also see May and Moore 2019). How the mismatch between the generating regime and the regime used for model estimation can affect the results?

Line 237: “caused by”

Lines 250 and 261: How these conclusions compare and contrast with the discussion of Caetano and Harmon (2019 – Systematic Biology).

---

## Round 0.2 · accepted · Accept

The authors have submitted a new improved version (the previous one received an editorial decision of minor revisions). The authors have replied or added the suggestions made by the reviewers. The current version is ready to be accepted and published in PeerJ.

·

Basic reporting

The authors have addressed all of my concerns and suggestions and where they have not they provide reasoning. I have no further comments and I recommend this manuscript for publication.

Experimental design

no comment

Validity of the findings

no comment